



# Impacts of changes in groundwater recharge on the isotopic composition and
# geochemistry of seasonally ice-covered lakes: insights for sustainable
# management
**Marie Arnoux[1,2], Florent Barbecot[1], Elisabeth Gibert-Brunet[2], John Gibson[3], Aurélie Noret[2]**
*[1] GEOTOP, Université du Québec à Montréal, Montréal, Québec, Canada H3C 3P8*
*[2] GEOPS, UMR 8148, CRNS-Université Paris Saclay/Paris-Sud, Orsay, France*
*[3] Alberta Innovates Technology Futures, 3-4476 Markham Street, Victoria, BC V8Z 7X8, Canada*
Keyword: lakes; climate change; recharge; groundwater-surface water interaction; hydrological balance; water stable
isotopes
**Corresponding author**:
Marie Arnoux (marie.arnoux@u-psud.fr)

17 +33 6 79547616

*GEOTOP, Université du Québec à Montréal, Montréal, Québec, Canada H3C 3P8*
*GEOPS, UMR 8148, CRNS- Université Paris-Saclay/Paris-Sud, Orsay, France*
Florent Barbecot (barbecot.florent@uqam.ca)
*GEOTOP, Université du Québec à Montréal, Montréal, Québec, Canada H3C 3P8*
Elisabeth Gibert-Brunet (elisabeth.gibert@u-psud.fr)
*GEOPS, UMR 8148, CRNS- Université Paris Saclay/Paris-Sud, Orsay, France*
John Gibson (jjgibson@uvic.ca)
*Alberta Innovates Technology Futures, 3-4476 Markham Street, Victoria, BC V8Z 7X8, Canada*
Aurélie Noret (aurelie.noret@u-psud.fr)
*GEOPS, UMR 8148, CRNS- Université Paris Saclay/Paris-Sud, Orsay, France*





## ABSTRACT

Lakes are under increasing pressure due to widespread anthropogenic impacts related to rapid development and population growth. Accordingly, many lakes are currently undergoing a systematic decline in water quality. Recent studies have highlighted that global warming and the subsequent change in water use may further exasperate eutrophication in lakes. Lake evolution depends strongly on hydrologic balance, and therefore on groundwater connectivity. Groundwater also influences the sensitivity of lacustrine ecosystems to climate and environmental changes, and governs their resilience. Improved characterization of groundwater exchange with lakes is needed today for lake preservation, lake restoration, and for sustainable management of lake water quality into the future.

Small groundwater-connected lakes were chosen to simulate changes in water balance and water quality expected under future climate change scenarios, namely Representative Concentration Pathways (RCP) 4.5 and 8.5. Contemporary baseline conditions, including isotope mass balance and geochemical characteristics, were determined through an intensive field-based research program prior to the simulations. Results highlight that future lake geochemistry and isotopic composition trends will depend on four main parameters: location (therefore climate conditions), lake catchment size (which impacts the intensity of the flux change), lake volume (which impacts the range of variation), and lake G-index (i.e., the percentage of groundwater that makes up total lake inflows), the latter being the dominant control on water balance conditions, as revealed by the sensitivity of lake isotopic composition. Based on these model simulations, stable isotopes appear to be especially useful for detecting changes in recharge to lakes with a G-index of between 50% and 80%, but response is non-linear. Simulated monthly trends reveal that evolution of annual lake isotopic composition can be dampened by opposing monthly recharge fluctuations. It is also shown that changes in water quality in groundwater-connected lakes depend significantly on lake location and on the intensity of recharge change.



## 1. INTRODUCTION


For decades, climate change, combined with rapidly expanding urban, industrial, and
agricultural water needs, has placed increasing stress on water resources and on groundwater
resources in particular. Future pressure on these resources is likely to be even more pronounced, as
groundwater is likely to be increasingly exploited to enhance water supply and to alleviate the worsening
drought situation in some arid regions (Dragoni and Sukhija, 2008). Many studies have suggested that
sustainable groundwater use has to be based on, among other things, a reliable assessment of
recharge, which largely controls its evolution. Aquifer recharge refers to the quantity of water reaching
the saturated zone of an aquifer, and therefore replenishing the water table. Unfortunately, in many
parts of the world, recharge rates are often not well-known at the regional scale (Rivard et al., 2013).
While aquifer recharge is crucial to supporting sustainable management of regional groundwater
resources, it is difficult to accurately estimate, owing mainly to limited data availability, as well as
limitations inherent to estimation methods and field measurements (Rivard et al., 2013). Recharge rates
are controlled by geology, soil characteristics, topography, land cover, land use and climate (Rivard et
al., 2014). Thorough literature reviews of the various techniques that exist to quantify groundwater
recharge are provided in Scanlon et al. (2002) and Healy (2011). Many methods can be used to
estimate groundwater recharge, such as water budget methods, modelling methods, tracer methods,
and methods based on surface water interaction studies. The latter is based on the estimation of
groundwater discharge to surface water, mainly by streambed seepage determination, stream flow
duration curves, or stream flow hydrograph separation (Scanlon et al., 2002). The recharge amount (in
mm.yr$^{-1}$) is then typically obtained by dividing measured or estimated discharge flow by the surface
drainage area at the measurement site. This procedure assumes that aquifer boundaries coincide with
watershed boundaries, and consequently that the area of the aquifer that contributes to groundwater
discharge is equal to the surface drainage area (Kuniansky, 1989; Rutledge, 1998, 2007). However, this
assumption must be considered carefully, as groundwater basins and watershed boundaries can differ
drastically (Tiedeman et al., 1997). Miscalculation of the aquifer contributing area will lead to a
proportional error in recharge estimate.
Although the discharge flow calculation method is commonly used to estimate recharge for
streams, it is less commonly used for surface water bodies, probably due to the greater difficulty of
quantifying groundwater discharge in these settings. However, in recent years some studies have
proven that groundwater flow into lakes can be reliably quantified. Interactions between lakes and
groundwater depend on geology, soil and sediment properties, and also on hydraulic gradient, which is





strongly dependent on climatic conditions and recharge (Winter, 1999). Therefore, variation in groundwater fluxes may indicate a change in recharge in the lake catchment (Meinikmann et al., 2013).

In Quebec (Canada), more than ten percent of the surface is covered by freshwater, with more than one million lakes known to exist. In many cases, these are connected to underlying aquifers. However, lake-groundwater interactions are highly dynamic throughout the year, and, even if it now possible to quantify groundwater inflow with a reasonable degree of confidence, it is difficult to determine how and to what extent lakes can be sensitive to changes in groundwater recharge. The lake water isotopic composition has been proven to be particularly useful for determining water balance parameter controls under changing conditions. For example, as shown in Turner et al. (2010), lake isotopic composition can highlight that (i) reduced winter precipitation could cause snowmelt-dominated lakes to become rainfall-dominated lakes, or that (ii) during longer ice-free seasons, mainly rainfall-dominated, but also potentially snowmelt-dominated lakes, may turn into evaporation-dominated lakes. Moreover, among all the methods used to quantify groundwater inflow to lakes, isotopic balances appear to be especially well-adapted for quantifying groundwater flux variations on seasonal and yearly time scales (Arnoux et al. 2017a). Water stable isotopes are therefore expected to be very useful for monitoring seasonal and inter-annual variations in the water budget under changing recharge conditions.

The impact of climate change on groundwater recharge is not easy to determine, because of the complexity of interactions and processes evolved, and can varies vastly depending on regions (Rivard et al. 2014; Crosbie et al., 2013). In addition, it is predicted to shift differentially under various climate scenarios and models (Jyrkama and Sykes, 2007; Levison et al., 2014). In Canada, highly variable recharge rates have been proposed in previous studies; for example, for the 2050 horizon (mainly the period 2041-2070) relative to modern (2000-2015) or past recharge rates (1950-2010), depending on study site, scenario, and model: +10 to +53% in the Grand River watershed, Ontario (Jyrkama and Sykes, 2007), -41 to +15% in the Chateauguay River watershed, Quebec (Croteau et al., 2010), –6 to +58% in the Otter Brook watershed, New Brunswick (Kurylyk and MacQuarrie, 2013), -4 to +15% at Covey Hill, Quebec (Levison et al., 2014), +14 to +45% in the Annapolis Valley, Nova Scotia (Rivard et al., 2014), and -28 to +18% for the Magdalen Islands, Quebec (Lemieux et al., 2015).

Recharge fluctuations can also impact lake water quality by changing groundwater fluxes, which are closely linked to phosphorous (P) loading to lakes. It is known that lake water quality is mainly driven by variations in P load, since this plays a critical role in limiting lake primary productivity and algal biomass, which in turn regulate lake trophic status. Increasing P concentration in the water column is the primary factor responsible for accelerated eutrophication and associated algae blooms (Schindler, 1977; Wang et al., 2008). At sites without urban drainage or point P sources, such as sewage treatment





plants, domestic waste from septic systems may represent the largest anthropogenic source of P to
lakes on the Canadian Shield (Dillon and Evans, 1993). Increases in shoreline development and
population, combined with groundwater fluxes variations, can clearly impact lake quality, but still remain
to be quantified.
For the present study, ten lakes in southern Quebec were sampled to quantify their yearly
groundwater inflows (see Arnoux et al., 2017a for more details), and one of these lakes was sampled
over the course of a year to quantify its monthly groundwater inflows (see Arnoux et al., 2017b for more
details). The main objectives of this study were (i) to determine how future groundwater recharge
changes might affect lake water balance and geochemistry, and (ii) to assess whether stable isotopes
might be an effective tool for identifying lakes that are susceptible to change or are undergoing changes
in water balance and water quality. To address these objectives, seasonal models of water and isotopic
budgets were established for several lakes, and the models were then forced with future yearly and
monthly time scale climate data from predictive global models to simulate anticipated conditions.
Climate outputs of the Canadian Regional Climate Model were used, based on scenarios RCP 4.5 and
RCP 8.5 (Moss et al., 2010; IPCC, 2014). It is assumed that recharge fluctuation is the main parameter
influencing groundwater fluxes into lakes, and thus a percentage of recharge change will lead to the
same percentage of change of groundwater fluxes to lakes. Different recharge scenarios, which
translate into changes in groundwater inflow, were then tested to determine changes in water budget
and isotopic evolution of the lakes. Predicted changes in recharge were then compared to predicted
population growth in the study areas to discuss lake quality evolution. After determining the evolution of
the lake geochemical signature, how lakes connected to groundwater can be used to identify changes in
groundwater recharge can be determined, as can whether or not the isotopic composition of lakes can
serve as an effective indicator of change or variability.



2. METHOD
2.1. Study sites
The ten lakes chosen are located in four regions of southern Quebec characterized by
contrasting climatic conditions: Laurentides (LAU), Outaouais (OUT), Abitibi-Témiscamingue (AT), and
Saguenay-Lac –Saint-Jean (SAG). These kettle lakes, set in coarse-grained (sand/gravel) fluvioglacial
deposits, are specifically targeted in this study, because they (i) are small enough to be sensitive to
environmental changes on a short time scale, (ii) do not have permanent surface inflow streams, and so
are largely groundwater dependent, (iii) are generally characterized by predictable and uniform
geomorphological features, and (iv) are likely connected to shallow, unconfined aquifers (Arnoux et al.
2017a; Isokangas et al., 2015). Kettle lakes originate as depressions in the landscape formed following
the melting of ice blocks buried in the ground after glacial retreat of the Late Glacial to Holocene
transition period (from -12 to -7 kyr). These kettle holes, becoming kettle lakes when they are filled with
water, are mainly found in fluvioglacial deposits, such as outwash plains, deltas, eskers, and kame
terraces (Benn and Evans, 2011). Figure 1 shows the locations of the ten lakes analyzed here. Their
main characteristics are described in Table 1.


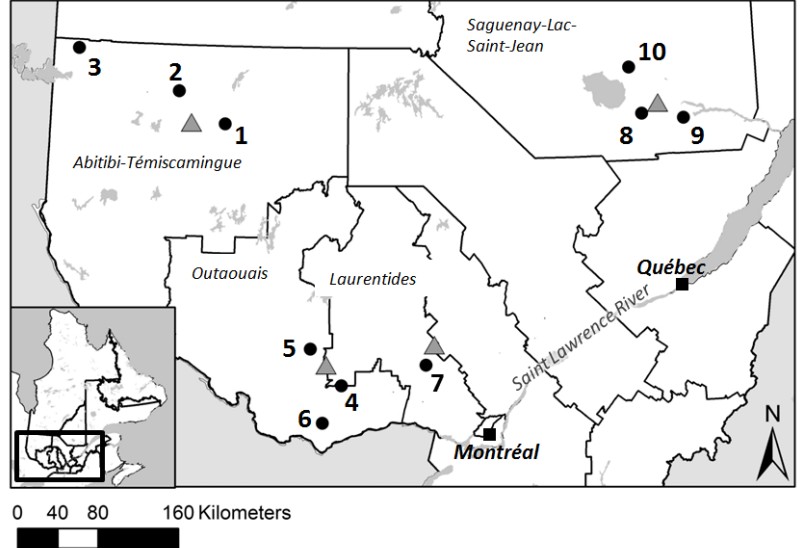

Fig. 1. Locations of the study lakes (circles) and sources of climate data (triangles)






## 2.2. Lake isotopic composition
### 2.2.1. Sampling
Water samples from each lake were retrieved during two field campaigns, in June-July and
October-November 2014. When physicochemical parameters, measured *in situ* along the water column,
revealed a well-mixed lake, the lake was considered to be homogeneous, and only one sample was
collected, from close to the lake bottom, at its greatest depth. Otherwise, for stratified periods, two
samples were collected: one from the top of the epilimnion and one from the base of the hypolimnion, in
order to obtain the complete range of isotopic composition variation. Whenever possible, groundwater
was sampled from private wells located in the vicinity of the studied lakes. Untreated groundwater
samples were collected from residential wells from the tap after purging approximately three times the
well volume.
Samples were transported in a cooler, and subsequently stored at 5°C until analyses were
performed. Water stable isotopic compositions were measured with a Laser Water Isotope Analyser (OA
ICOS DLT, Los Gatos Research, now ABB) at the GEOPS Laboratory (University of Paris-Sud/Paris-
Saclay, France). The measurement accuracy is $\pm$ 1 ‰ for $\delta^2H$ and $\pm$ 0.2 ‰ for $\delta^{18}O$. Results are
reported in δ values, representing deviations in per mil (‰) from the isotopic composition of the
international standard (Vienna Standard Mean Ocean Water, VSMOW), such that $\delta^2H$ or
$\delta^{18}O=((R_{sample}/R_{VSMOW})-1)\times1000$, where R refers to $^2H/H$ or $^{18}O/^{16}O$ ratios.
One of the lakes, Lake Lacasse, was sampled in more detail throughout 2015-2016. Water
samples were collected from the lake at two week to one month intervals, mainly from the deepest part
of the lake, and at 1 to 2 meter depth intervals in order to monitor the vertical heterogeneity of the water
column. Groundwater was sampled twice from eight private wells in the vicinity of the lake (see Arnoux
et al, 2017b for more detail).

### 2.2.2. Water mass balance
The lake water budget is defined as:
$$\frac{dV}{dt} = I - E - Q \quad \text{Eq. (1)}$$

where V is the volume of the lake ($m^3$); t is time (days); E is evaporation ($m^3.day^{-1}$); I is the
instantaneous inflow ($m^3.day^{-1}$), corresponding to the sum of upstream surface inflow ($I_S$; zero for the
studied lake), runoff ($I_R$; considered negligible), groundwater inflow ($I_G$), and precipitation on the lake
surface (P); Q is the outflow ($m^3.day^{-1}$), which is the sum of surface ($Q_S$) and groundwater ($Q_G$) outflows.





Under constant atmospheric and hydrologic conditions, steady state is assumed (Gibson et al., 2016),
implying that dV/dt=0 and $I_G$ is therefore equal to $Q_S+Q_G+E-P$ for the entire lake.

2.2.3. Stable isotopic mass balance

Considering water stable isotopes, the lake isotopic mass balance is:

$$V\frac{d\delta_L}{dt} + \delta_L\frac{dV}{dt} = I\delta_I - E\delta_E - Q\delta_Q \quad \text{Eq. (2)}$$

where δ is isotopic composition of: the lake ($\delta_L$), total inflow ($\delta_I$), which include runoff ($\delta_R$), precipitation
($\delta_P$), surface inflow ($\delta_S$) and groundwater inflow ($\delta_G$), and total outflow ($\delta_Q$), which include surface ($\delta_{QS}$)
and groundwater ($\delta_{QG}$) outflows. The isotopic composition of evaporating water ($\delta_E$) was estimated
using the Craig and Gordon (1965) model, expressed by Gonfiantini (1986) as:
$$\delta_E = \frac{(\delta_L - \varepsilon^+)/\alpha^+ - h\delta_A - \varepsilon_K}{1 - h + 10^{-3}\varepsilon_K} \quad \text{Eq. (3)}$$

where h is the relative humidity at the lake surface; $\delta_A$ is the local isotopic composition of the
atmospheric moisture (‰); $\varepsilon^+ = (\alpha^+ - 1)*1000$ is the equilibrium isotopic separation (‰); $\alpha^+$ is the
equilibrium isotopic fractionation, and $\varepsilon_K = C_K(1-h)$ is the kinetic isotopic separation (‰), with $C_K$ being
the ratio of molecular diffusivities between heavy and light molecules (Gibson et al., 2016).

In this study, $C_K$ values were considered to be representative of fully turbulent wind conditions

and a rough surface for both oxygen ($C_K$ =14.2‰) and hydrogen ($C_K$ =12.5‰), based on experimental
data (Horita et al., 2008). For calculating equilibrium fractionation factors, experimental values of Horita
and Wesolowski (1994) were used:
$\alpha^+(^{18}O) = \exp(-7.685/10^3 + 6.7123/T - 1666.4/T^2 + 350410/T^3)$ Eq. (4)
$\alpha^+(^2H) = \exp(1158.8 \times T^3/10^{12} - 1620.1 \times T^2/10^9 + 794.84 \times T/10^6 - 161.04/10^3 + 2999200/T^3)$ Eq. (5)
where T is temperature (K). The isotopic composition of atmospheric moisture ($\delta_A$, ‰) was calculated
assuming equilibrium isotopic exchange between precipitation and vapor:
$$\delta_A = \frac{\delta_P - \varepsilon^+}{1 + 10^{-3}\varepsilon^+} \quad \text{Eq. (6)}$$

where $\delta_P$ (‰) is the mean annual isotopic composition of precipitation. Assuming well-mixed conditions
in the lake, the combination of Eq. (3) and Eq. (2) yields:
$$V\frac{d\delta_L}{dt} + \delta_L\frac{dV}{dt} = P\delta_P + I_G\delta_G - Q\delta_L - \frac{E}{1 - h + 10^{-3}\varepsilon_K}\left(\frac{\delta_L - \varepsilon^+}{\alpha^+} - h\delta_A - \varepsilon_K\right) \quad \text{Eq. (7)}$$

A steady state was assumed, such that dV/dt=0. Equation (7) can therefore be simplified to:




$$V\frac{d\delta_L}{dt} = P\delta_P + I_G\delta_G - (P + I_G - E)\delta_L - \frac{E}{1 - h + 10^{-3}\varepsilon_K}\left(\frac{\delta_L - \varepsilon^+}{\alpha^+} - h\delta_A - \varepsilon_K\right)$$ Eq. (8)

Resolving this calculation therefore allows isotopic composition of the lake water at time t+dt to be
determined, expressed as a function of its value at the previous time step, t, and two established
parameters, A (‰.m$^3$) and B (m$^3$):
$$\delta_L^{t+dt} = \frac{A}{B} + (\delta_L^t - \frac{A}{B})\exp(-\frac{B}{V}dt)$$ Eq. (9)

with
$$A = P\delta_P + I_G\delta_G - \frac{E}{1 - h + 10^{-3}\varepsilon_K}\left(-h\delta_A - \varepsilon_K - \varepsilon^+/\alpha^+\right)$$ Eq. (10)

$$B = P + I_G - E\left(1 - \frac{1}{\alpha^+(1 - h + 10^{-3}\varepsilon_K)}\right)$$ Eq. (11)

The monthly mean isotopic composition of precipitation ($\delta_P$) was assessed in the four regions
from the Global Network of Isotopes in Precipitation (GNIP) and Program for Groundwater Knowledge
Acquisition (PACES) datasets. Future $\delta_P$ trends are uncertain; however, they have been shown to be
mainly dependent on temperature evolution and local factors (Stumpp et al., 2014), and a recent study
in Siberia showed that a long term increase in precipitation $\delta^{18}O$ is close to the detection limit of the
tracers (<1‰ per 50 years) (Butzin et al., 2014). Monthly current means were therefore used in the
current simulations. The mean value of groundwater isotopic composition ($\delta_{Gi}$) was determined from the
mean groundwater isotopic composition measured in wells, located in the same region and presenting
no enrichment due to evaporation. The mean isotopic values used for groundwater are presented in
Table 2.

The uncertainties associated with the Craig and Gordon (1965) model in the estimated isotopic
composition of evaporating moisture ($\delta_E$) can be substantial, especially if relative humidity is greater
than 0.8 (Kumar and Nachiappan, 1999). Moreover, a sensitivity analysis of $^{18}O$ isotopic balance of a
small lake in Austria (Yehdegho et al., 1997) indicates that for flow-though, groundwater-dominated
systems with limited evaporation, the isotopic composition of the lake water and the inflow water are the
parameters critical to the overall uncertainty. Horita et al. (2008) recommended using time-averaged
values of the parameters in the calculation of $\delta_E$ for the given period of interest. Therefore, on an annual
time step, $\delta_P$ is monthly precipitation-flux weighted, except when it is used to estimate $\delta_A$; in this case,
$\delta_P$ is evaporation flux-weighted (Gibson, 2002; Gibson et al., 2016).
At a monthly time scale, monthly values are used for each parameter of the model, and
evaporation is considered to be null during the ice-covered period. Moreover, in winter, when monthly





mean temperature is below zero, precipitation is assumed to be zero in the model. Then, when monthly
temperature becomes equal to or higher than zero, accumulated precipitation and amount-weighted $\delta_P$
are added to the calculation during the melt period.

### 2.3. Evolution scenarios

#### 2.3.1. Climate models

In the present study, the fifth version of the Canadian RCM (CRCM5) was chosen, which has a
0.44° horizontal grid resolution (approx. 50 km; Sushama et al., 2010; Martynov et al., 2013; Šeparović
et al., 2013). The CRCM5 is a grid-point model, based on a two time-level, semi-Lagrangian, (quasi)
fully implicit time discretization scheme (Alexandru and Sushama, 2015). The model includes a terrain-
following vertical coordinate based on hydrostatic pressure (Laprise, 1991; Alexandru and Sushama,
2015), and an horizontal discretization on a rotated latitude-longitude, Arakawa C grid (Arakawa and
Lamb, 1977; Alexandru and Sushama, 2015). Following CRCM4, changes that have been introduced
into CRCM5 include, for example, evolution in the planetary boundary layer parameterization to
suppress both turbulent vertical fluxes under very stable conditions and the interactively coupled one-
dimensional lake model (Flake; Mironov et al., 2010; Martynov et al., 2012; Šeparović et al., 2013).
CRCM5 uses the Canadian Land-Surface Scheme (CLASS, version 3.5; Verseghy, 1991; Alexandru
and Sushama, 2015). This model is described in detail in Martynov et al. (2013) and Šeparović et al.

(2013).

The CRCMs were driven by the second-generation Canadian Earth System Model (CanESM2,
improved from CanESM1; Arora et al., 2011), developed by the Canadian Center for Climatic Modeling
and Analysis (CCCma). As explained in Šeparović et al. (2013), it consists of a fourth-generation
atmospheric general circulation model CanAM4, coupled with (i) the physical ocean component OGCM4
developed from the NCAR CSM Ocean Model (NCOM; Gent et al., 1998), (ii) the Canadian Model of
Ocean Carbon (CMOC; Christian et al., 2010), and (iii) Canadian Terrestrial Ecosystem Model (CTEM;
Arora and Boer, 2010). The CanAM4 is a spectral model employing T63 triangular truncation with
physical tendencies calculated on a 2.81 linear grid and 35 vertical levels (Arora et al., 2011; Šeparović
et al., 2013).

#### 2.3.2. Climate data

Four greenhouse gas concentration scenarios (Representative Concentration Pathways, RCP)
have been adopted by the IPCC in its fifth Assessment Report (AR5) in 2014: RCP 2.6, RCP 4.5, RCP
6.0, and RCP 8.5. The scenarios selected for the present study are RCP 4.5 and RCP 8.5, for which
predicted climate data are available until 2100 for the study regions. The RCP 4.5 scenario considers


that long-term global emissions of greenhouse gases and land-use-land-cover stabilize radiative forcing
at 4.5 $W.m^{-2}$ (approximately 650 ppm $CO_2$-equivalent) by the year 2100, without ever exceeding that
value. The RCP 8.5 scenario corresponds to the highest greenhouse gas emissions pathway scenario,
with gas emissions and $CO_2$ concentrations increasing considerably over time, and thus leading to a
radiative forcing of 8.5 $W.m^{-2}$ by the end of the century (approximately 1370 ppm $CO_2$ equivalent). The
defining characteristics of these scenarios are enumerated in Moss et al. (2010).

In order to connect these RCP forecasts to our study and to visualize trends, yearly mean data

are presented in Fig. 2. Based on previous literature on recharge changes (see part 2.2.3.), a reference
period (2010-2040) is compared to a future period (2041-2071). It is noted that both evaporation and
temperature display increases between the reference and future periods for both scenarios, although it
is more pronounced for RCP 8.5. Moreover, precipitation and relative humidity do not show clear trends.
However, it seems that precipitation variability will increase overall for both scenarios, although this is
more pronounced for RCP 8.5. Moreover, the southern regions (i.e., OUT and LAU) have higher
temperatures than the northern regions (i.e., AT and SAG), and precipitation is higher in LAU than in the
other three regions. On a monthly time scale, surface temperatures in LAU show an increasing monthly
trend, whereas evaporation increases mainly during summer and stays relatively constant the rest of the
year (data not shown). Meanwhile, precipitation does not show any clear trend. However, as
temperatures increases in winter, melt periods likely will shift more frequently occur earlier in the year.



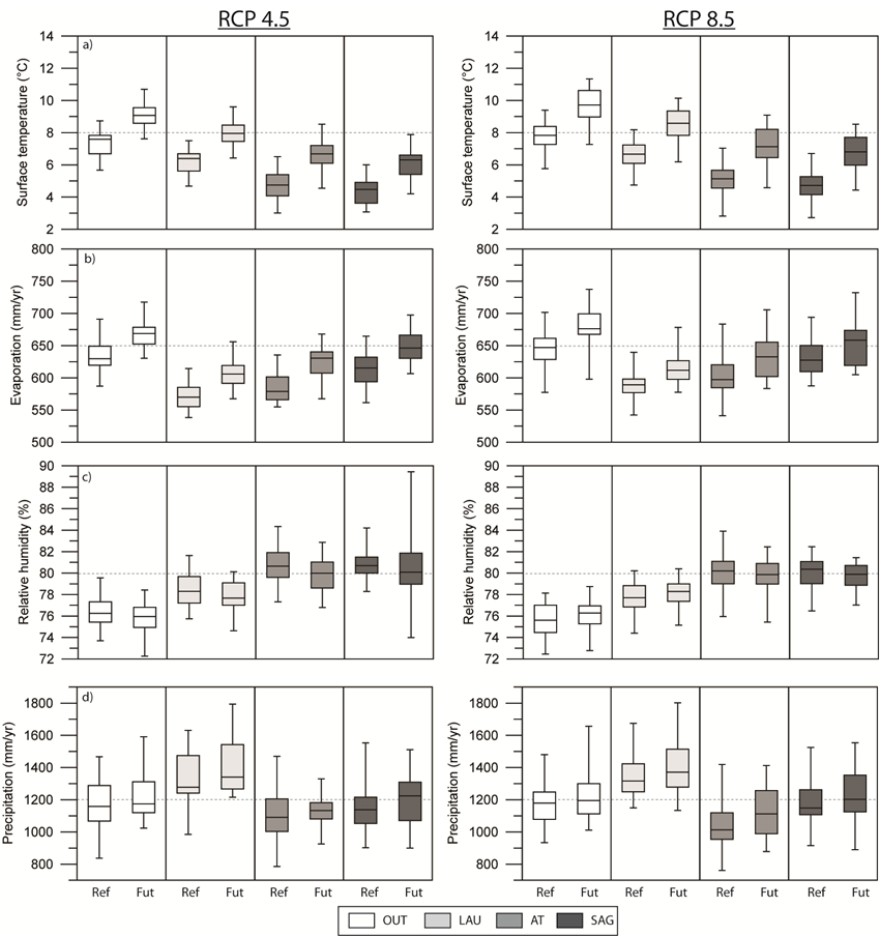


Fig. 2. Climate data for the reference (Ref; 2010-2040) and future (Fut; 2041-2071) periods, obtained from CRCM5 –CanESM2, with RCP
4.5 (left) and RCP 8.5 (right) scenarios for the four different study areas. The variables are: a) surface air temperature, b) surface water
evaporation (obtained from surface heat flux), c) surface relative humidity (obtained from surface specific humidity), and d) precipitation
(Martynov et al., 2013; Šeparović et al., 2013).





### 2.3.3. Recharge evolution

The mean annual recharge for each lake basin was obtained by dividing the lake drainage area

by the calculated mean annual groundwater inflow to the lake (Meinikmann et al., 2013). In this study,
recharge evolution is thus expressed in terms of changes in groundwater inflow to the lakes.

In the first step, recharge is assumed to be constant for the 2006-2014 period. Over this period,

recharge is adjusted to fit the calculated lake isotopic compositions to those measured. In the second
step, the results of Rivard et al. (2014) was chosen for the simulation of recharge scenarios, since this
study focusses on the Annapolis Valley (Nova Scotia, Canada), not far from southern Quebec and with
a similar latitude, geology, and climate. Therefore, the future recharge dynamics determined for the
Annapolis valley are assumed to be similar to those of the present study sites. Rivard et al. (2014) found
that all scenarios predict an annual recharge to the aquifer within the range of +14 to +45% higher than
at present by 2041-2071. They also predict, on a seasonal basis, that recharge will undergo (i) a marked
decrease in summer (from 4 to 33%), and (ii) a spectacular increase in winter (more than 200%), due to
an earlier melt period starting date.

The following section focussed firstly on monthly lake isotopic composition evolution (Part 3.1.)

and secondly on yearly lake isotopic composition evolution (Part 3.2.). Monthly and yearly values are
compared for the two standard periods (i.e., for reference (2010-2040) and future (2041-2071) periods).

For the first part of the study, Lake Lacasse, located in the LAU region, has been chosen, since

it was subject to continuous monitoring (Arnoux et al., 2017b). Its groundwater inflow and variability has
therefore already been well-constrained throughout the year 2015-2016 (Fig. 3 b). For this lake, the
model was run from 2006 to 2071, and four different recharge evolution scenarios were applied to the
2041-2071 period, following the predictions of Rivard et al., 2014 for scenarios S1 and S2, as described
below.
- NC: no change in recharge (groundwater inflow follows the pattern described in Fig. 3,

obtained from Arnoux et al., 2017b);

- S0: a recharge decrease of 33% during the summer period (from June to October);
- S1: a 200 % increase in recharge during the melt period (from January to March), and a 4%

decrease in the summer period;

- S2: a 200 % increase in recharge during the melt period, and a 33% decrease during the

summer period.

For the second part, three annual recharge evolution scenarios were tested, following the

predictions of Rivard et al., 2014: no change (NC), a 14% increase (Low), and a 45% increase (High) in
mean annual recharge.





### 2.4. Population growth


Variations in the quantity and/or quality of groundwater feeding lakes can obviously impact the
geochemistry, and thus the water quality of lakes, especially for lakes displaying a high G-index (the
percentage of groundwater comprising the total lake inflow; Arnoux et al., 2017a). Moreover, in rural
areas of Quebec, lake and groundwater quality is likely to be influenced by changes in population
density. The population of Quebec is aging, and many seasonal residences (e.g., cottages) around
lakes in rural areas are expected to become year-round residences. Furthermore, these residences are
not connected to waste water treatment plants; rather, owners have their own private wells for drinking
water and private septic tanks with subsurface seepage beds for waste water. The predicted population
changes are summarized in Table 3. Population is mainly expected to increase in the southern regions
(OUT and LAU), with a mean increase of 24 and 28% respectively (ISQ, 2014; Table 3). Scenarios of
population growth are compared with scenarios of recharge evolution for each lake to assess their
future quality evolution.

## 3. RESULTS AND DISCUSSION


### 3.1. Monthly evolution of lake isotopic composition


Figure 3 shows the measured and modelled isotopic compositions of Lake Lacasse. It can be
observed that the modelled values are more variable than the measured ones, undoubtedly due to the
higher evaporation rate in the climatic model (459 mm) than that measured during the field monitoring
period (204 mm). It is also shown that the model attributes greater weight to the contribution of the
depleted snow value of the than is realistic. This is probably due to the snow column (which is close to
0°C during the snow melt) being less dense than the lake surface water (which has a mean temperature
of close to 4°C), and therefore bypasses the lake, flowing rapidly out of the lake outlet. In such a case,
the snow does not influence the lake isotopic composition as much as the model predicts. Since similar
results are obtained for $\delta^2H$ values, only the $\delta^{18}O$ results from the model will be presented in the
following sections.

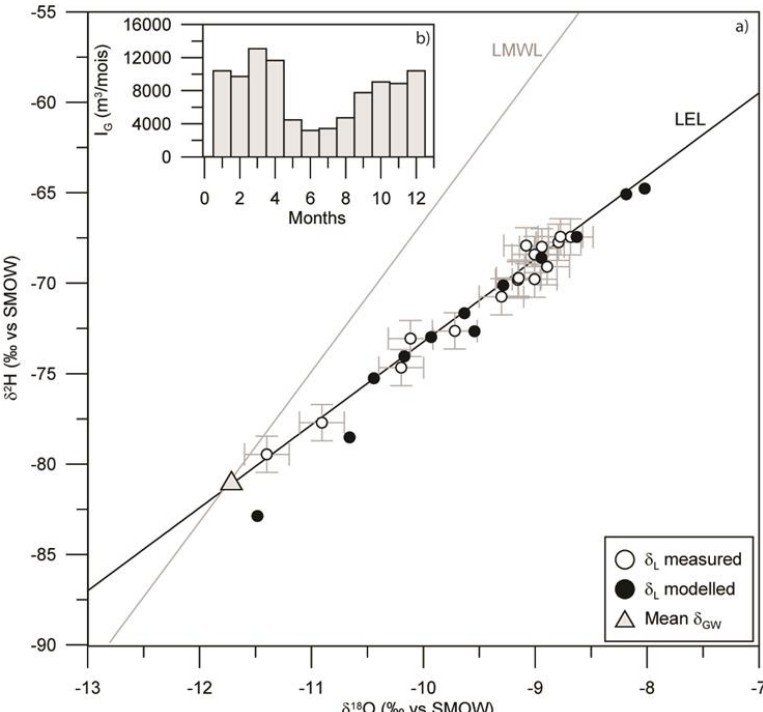

Fig. 3. (a) Isotopic composition of Lake Lacasse between June 2015 and May 2016, measured and modelled following; (b) the pattern of groundwater inflow ($I_G$) to Lake Lacasse.

Lake Lacasse has a mean G-index of 69% during the reference period. Results for monthly simulations, with RCP 4.5 climate data, are illustrated in Fig. 4. Lake isotopic compositions are not significantly different between the reference and future periods if no change is applied to the recharge pattern (Fig. 4). Under scenarios S1 and S2, it can be observed that future $\delta^{18}O$ is nearly 100% different from reference conditions during the two first months of the year (Fig. 4). It is at least 75% different for the month of March, but this month shows important variation during the future period. Throughout the rest of the year, ranges of variation are not completely different, but increasing or decreasing trends can be observed, depending on the season.

Indeed, Fig. 5 shows the monthly differences between mean lake $\delta^{18}O$ in the reference period and mean lake $\delta^{18}O$ in the future period, for the four recharge evolution scenarios. The highest variation in the reference period is observed in March for S1 (-1 ‰), S2 (-1 ‰), and NC (-0.4 ‰), after the melt period. For S0, the greatest change during the reference period is observed in September and October (+0.4 ‰), after the evaporation period. For the future period, the greatest difference between winter recharge is in February (-0.6 and -0.5 ‰ for S1 and S2 respectively). This suggests that future changes in lake isotopic composition associated with recharge may be highest in February.

As presented in Fig. 5, during the summer, flow variation can be observed:



-      Regarding the reference period: the highest variation will be in August for NC (+0.2 ‰),
while it will be in September and October for S0 (+0.4% for both months) and S2 (+0.2 and
+0.3 ‰ in September and October respectively). S1 do not show any variation.

-      Regarding the NC future period: the greatest change will be in October for S0 (+0.3 ‰) and
S2 (+0.2 ‰), and in September for S1 (-0.1 ‰).


Results of scenario S2, characterized by the greatest changes in recharge, in both summer and
winter, highlights that the impact of decreased recharge during summer attenuates the substantial
impact of increased recharge during winter. Indeed, during winter, S1 shows more depleted values than
S2 (-0.5 versus -0.4 ‰ in January, and -0.8 and -0.7 ‰ with respect to the reference period for S1 and
S2 respectively). Therefore, the more recharge decreases in the summer, the more lake isotopic
composition increases in the summer, due to increased future evaporation. Meanwhile, the more
recharge increases in the winter, the more lake isotopic composition is depleted in the winter. If both
phenomena occur in a given year, the mean annual lake isotopic composition evolution will therefore not
be expected to shift much, since their opposing impacts on lake isotopic composition will cancel each
other out. As such,  S1 is the scenario showing the highest variation in annual mean, of -3 ‰, compared
with -2 ‰ for S2 and +2‰ for S0.

Based on these observations, it appears that isotopic signatures measured at the end of
February and in September or October will provide information on the greatest changes during the
winter and summer periods respectively. The greatest changes in lake isotopic composition are likely to
be at the end of the melt period.



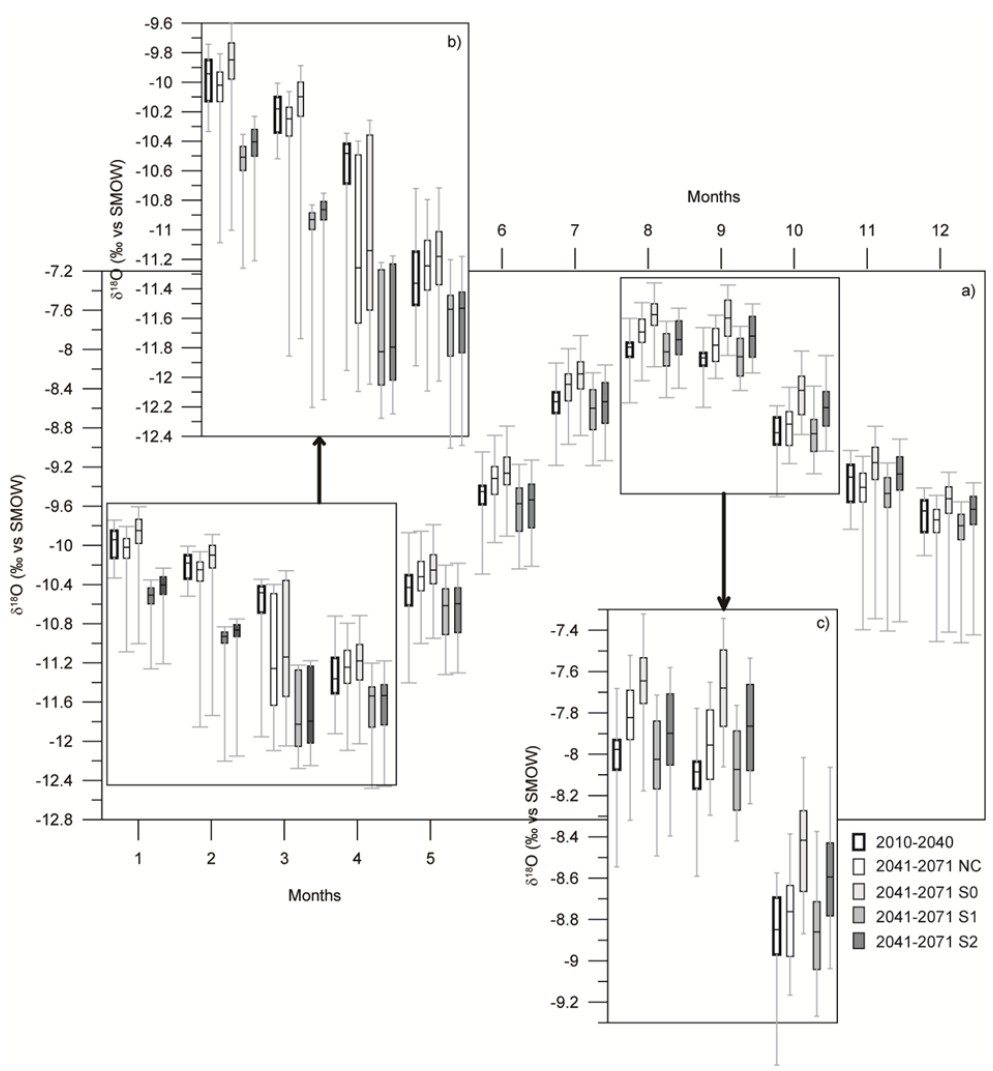


Fig. 4. (a) Monthly Lake Lacasse isotopic composition, calculated using RCP 4.5 climatic data, for different periods and various recharge
patterns: no change (NC), -33% in the summer (from June to October; S0), +200 % during the melt period (from January to March) and -
4% in the summer (S1), and +200 % during the melt period and -33% in the summer (S2); (b) close-up of the winter months; c) close-up of
the summer months.



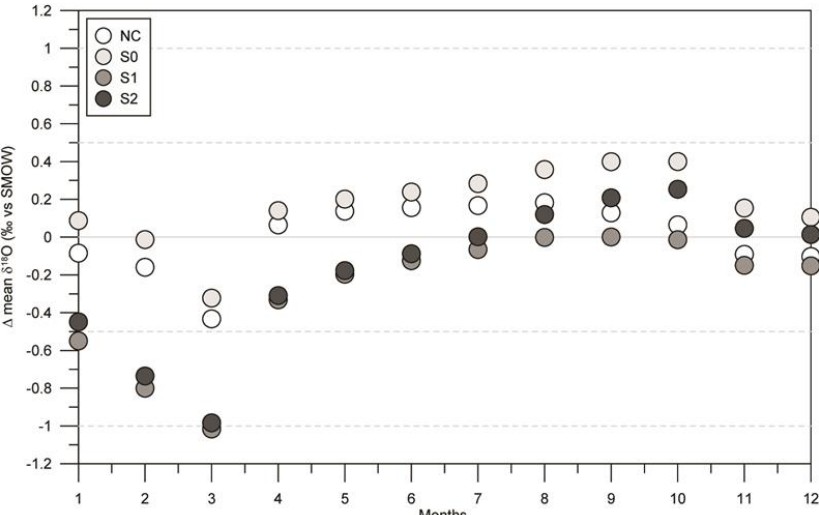

Fig. 5. Differences between mean Lake Lacasse δ18O in the reference period and mean Lake Lacasse δ18O in the future period, for the RCP 4.5 climate scenario and four scenarios of recharge evolution: no change (NC), -33% in the summer (from June to October; S0), +200 % during the melt period (from January to March) and -4% in the summer (S1), and +200 % during the melt period and -33% in the summer (S2).

Moreover, simulation results show that RCP 4.5 and 8.5 models provide similar results for Lake Lacasse isotopic composition evolution. Figure 6 shows the comparison of lake δ18O composition for both RCP climate scenarios, from 2010 to 2071, assuming the NC recharge scenario. In Fig. 6, it can be observed that there is a small trend toward δ18O enrichment due to a higher evaporation rate, which is more pronounced for the RCP 8.5 than for the RCP 4.5 scenario. However, on a yearly time scale, the impact of evaporation increase in the summer seems to be attenuated by a precipitation increase throughout the rest of the year, likely implying that these climate changes result in a nearly non-measurable impact on lake isotopic composition evolution.

Finally, all these results show that extreme caution is required when interpreting trends in lake isotopic composition, and that their interpretation requires (i) a minimum background knowledge – at least one year of data – of lake isotopic composition evolution in relation to its hydrological balance, and (ii) an accurate evaluation of weather data variability in the year of monitoring, with respect to their annual means for the study lake. A long term change in recharge will definitely impact lake isotopic composition, but the lake is also sensitive to changes in other water budget parameters. It may therefore still be difficult to definitively isolate the effect of recharge over long time periods. As such, it is also important to consider evolution in the yearly mean lake isotopic composition.




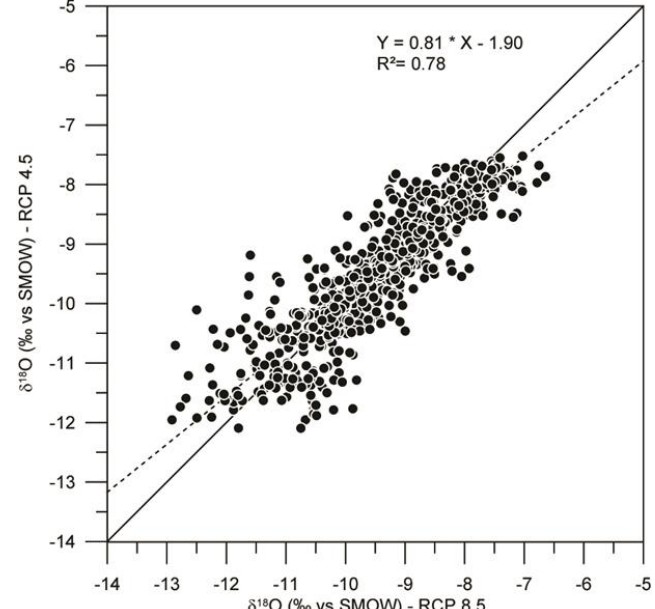

Fig. 6. Comparison between monthly results in δ¹⁸O for both scenarios RCP 4.5 and 8.5 for the 2010-2171 period.

## 3.2. Annual evolution of lake geochemistry
### 3.2.1. Isotopic signature evolution
The model was run for the ten study lakes, including Lake Lacasse (Table 1 for main lake
characteristics). Figure 7 illustrates differences in $\delta^{18}O$ in the reference period compared to the future
period for lakes which have a range of G-indices. It can be observed that, if the recharge is set as
constant from 2010 to 2071 (NC recharge scenario), there is no significant difference between the
reference and future period (Fig. 7), although evaporation shows a significant increase with time. The
lack of a trend is probably mitigated by concurrent shifts in precipitation (Fig. 2). Without considering
changes in groundwater inflow, it appears that lake isotopic composition will be at least as much
impacted by changes in precipitation as by changes in evaporation.
Fig. 7 illustrates that the range of lake isotopic composition variation depends significantly on
climate conditions, lake volumes, and their associated G-indices. It can be observed that lakes with a
low G-index and a small volume have higher potential variability in isotopic composition than those with
a high G-index and high volume. For example, for two lakes with a similar mean G-index, such as Lake
Ludovic (SAG; G-index=51%) and Lake Lacroix (OUT; G-index=53%), the former is expected to have a
greater spread in isotopic compositions than the latter, even though the SAG region will likely undergo





less evaporation increase compared with the OUT region (Fig. 2). This difference is due to the lower
volume of Lake Ludovic (V=400*10$^3$ m$^3$), compared with Lake Lacroix (V=1080*10$^3$ m$^3$; Table 1). In
addition, when lakes have a high G-index, the groundwater flux tends to buffer lake isotopic variations,
and so they tend to be less sensitive to changes in climate data. The dominant control on lake isotopic
variability therefore appears to be the G-index. Another example is Lake Lanthier, which has a smaller
volume (V=125*10$^3$ m$^3$) and a higher G-index (G-index=94%), and therefore shows a limited range of
isotopic variation compared with Lake Lacroix, although both are located in the OUT region (Fig. 7).

If a changing recharge scenario is applied, a decreasing trend in lake isotopic composition is

clearly observed (Fig. 7). However, it is also shown that lakes are sensitive to large changes in annual
recharge (+45%), but the differences are not significant if a smaller change (+14%) occurs. Moreover,
as the percentage of recharge change applied in the model is the same for all lakes, it can be observed
that the trend intensity will depend on four main parameters: lake catchment size (which controls the
intensity of the flux change), the region (which underlies climate condition), lake volume (which impacts
the range of variation), and the G-index. However, a significant relationship is only found with the latter.

Figure 8 illustrates variations in mean lake δ$^{18}$O versus G-index in both reference and future

periods. As shown, lake isotopic composition is more sensitive to changes in recharge for G-indices
ranging from 50 to 80%, with a maximum of sensitivity observed for a G-index of around 65 %. It can
also be observed that RCP 8.5 predicts a more depleted isotopic composition than does RCP 4.5. This
implies that for the same recharge scenario, variations in precipitation and melt period (duration and
time in the year) may impact the lake isotopic evolution more than precipitation. Finally, the polynomial
relationship between the two variables in Fig. 8 highlights that the G-index drives the response of lake
isotopic composition to changes in recharge.







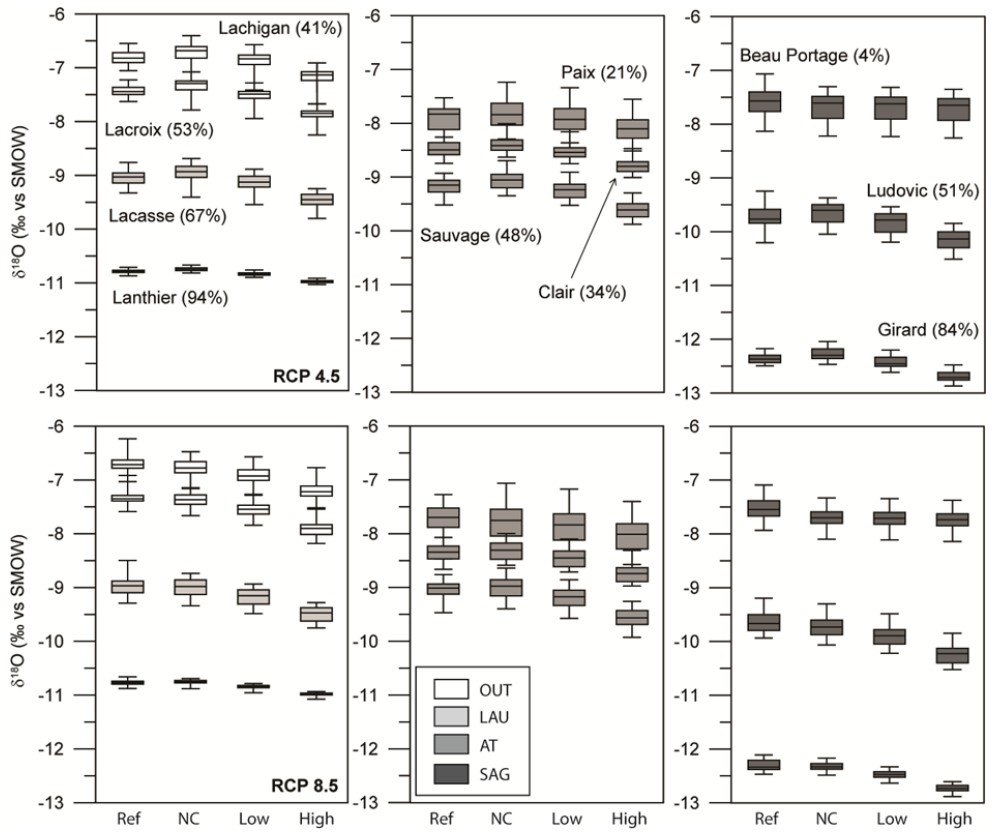

Fig. 7. Reference period (Ref; 2010-2040) lake δ$^{18}$O composition and that corresponding to three different future period (2041-2071) recharge scenarios: no change (NC), +14% (Low), and +45% (High), for RCP 4.5 (top) and RCP 8.5 (bottom) scenarios. The values in brackets correspond to the mean G-index (percentage of groundwater flow in the total inflow) for each lake calculated for the reference period; left panels show OUT and LAU regions, middle panels AT and rights panels SAG.

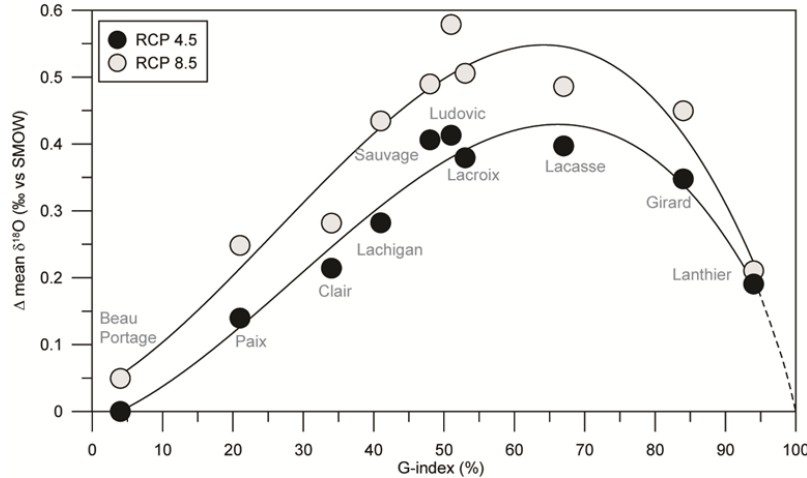

Fig. 8. Differences between mean lake δ$^{18}$O in the reference period (2010-2040) and future period (2041-2071), for the higher recharge change scenario, versus lake G-indices. RCP 4.5 (black dots) and 8.5 (grey dots) scenarios are represented.





### 3.2.2. Lake quality evolution

Turning to the predictions of population growth summarized in Table 3, population is predicted to increase mainly in the southern regions, OUT and LAU, with a mean increase of 24 and 28% by 2036 respectively (ISQ, 2014). Assuming an identical per capita P load, total P load in groundwater originating from waste water should increase by the same percentage.

Domestic sewage is the main contribution of anthropogenic sources to the total P load for most of Canadian lakes (Dillon and Evans, 1993; Paterson et al., 2006). The total P load from sewage systems is a function of (i) the population and (ii) the annual P consumption per capita (Paterson et al., 2006). As done by Paterson et al. (2006), assuming an effluent concentration of 9 mg.L$^{-1}$ (considering reductions in the phosphate content of detergents) and a daily water usage of 200 L.capita$^{-1}$.day$^{-1}$, the P contribution is estimated to be 0.66 kg.capita$^{-1}$.yr$^{-1}$. Investigated lakes in the OUT and LAU regions collect sewage from 4 (Lake Lachigan), 53 (Lake Lanthier), 117 (Lake Lacroix), and 17 houses (Lake Lacasse) within their catchments respectively. If two habitants per house are assumed, P loading to groundwater will be increased from 1 to 39 kg.yr$^{-1}$ in the studied lakes in these areas.

The impact of this P load increase on lakes can then roughly be estimated based on the ratio of change in annual P load versus change in annual recharge, as illustrated in Fig. 9. For an increase in recharge, if $\Delta_P/\Delta_R < 1$, the change in recharge over the catchment, and thus the evolution of the groundwater inflow to the lakes, will greater than the P variation. In such a case, the lake water quality may not be impacted by this P variation. On the other hand, if $\Delta_P/\Delta_R > 1$, the lake water quality will be impacted, and precaution should be taken to minimize the risk of blue-green algae blooms and consequent eutrophication. For the study regions (Fig. 9), if recharge increases 14% by 2036, as estimated by Rivard et al., 2014, lakes in the LAU and OUT areas will experience a decrease in their water quality. However, if the recharge change is closer to +45% (Rivard et al., 2014), lake water quality should not be worse than today, providing all other things remain equal and assuming the population growth forecasts are accurate.


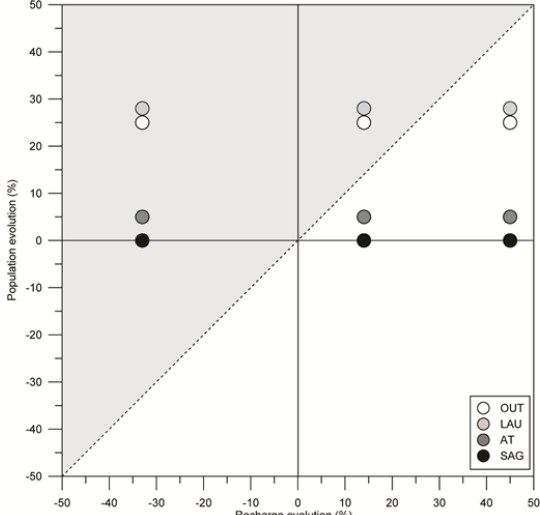


Fig. 9. Population growth prediction versus change in recharge. The shaded area represents the scenarios for which lakes may be under
risk of too high P loading, and therefore at risk of a decrease in water quality.

## 4. CONCLUSION

The main objectives of this study were to determine how future trends groundwater recharge
can affect lake geochemistry, and to assess whether stable isotopes might be an effective tool for
identifying lakes that are susceptible to change, or are undergoing changes, in their water budget and
quality.
Firstly, climate predictions from both RCPs 4.5 and 8.5 scenarios and their impacts on future
lake isotopic composition have been considered. By 2050, temperature and evaporation are expected to
increase, and precipitation to exhibit a slightly increasing trend, all trends being more intense under the
RCP 8.5 scenario. On a monthly time step, it has been highlighted that future lake isotopic signatures
will be more depleted with respect to the reference period, mainly in March and February, because of an
earlier melt period. In the summer, lake isotopic composition will be more enriched, mainly in August,
due to the higher evaporation rate expected. However, future variations with respect to the reference
period are smaller in the summer than in the winter. Scenario RCP 8.5 induces more intense monthly
variations, but no significant difference in future lake isotopic signatures is observed on a yearly time
step between the two scenarios. This means that enrichment caused by increased evaporation
compensates for depletion induced by precipitation variation. It is therefore unclear whether lakes will be
impacted more by increased evaporation or precipitation changes. Caution is therefore recommended in
the interpretation of isotopic trends in lakes where background knowledge – for at least one year – of
their isotopic composition evolution with respect to weather data and their hydrologic balance is lacking.




It has then been demonstrated that future lake isotopic composition will also depend on
recharge fluctuations, in addition to climate conditions. On a monthly basis, the highest impact of
recharge evolution on future lake isotopic composition will be in February. Moreover, if recharge
decreases during the summer, the main difference will be observed at the end of the summer, after the
evaporation period and before recharge stops decreasing, in September or October. Therefore, to
clearly identify future changes in recharge through the lake isotopic signature evolution, sampling only at
the end of February and in September or October will provide information on the greatest changes for
the winter and summer periods respectively.

On an annual time step, modelled evolutions of lake isotopic composition can clearly be
sensitive to both +45% and +14% changes in recharge, less so, nevertheless, to the latter. The intensity
of the future trend of lake isotopic composition will depend on four main parameters: lake catchment
(which controls the intensity of the flux change), the region (which drives climate conditions), lake
volume (which impacts the range of variation), and the G-index (which is the dominant control on water
balance conditions). Based on these model simulations, stable isotopes appear to be especially useful
for detecting changes in recharge to lakes with a G-index of between 50% and 80%.

It is important to keep in mind that if both a winter increase and summer decrease in recharge
occur during the same year, the trend in mean annual lake isotopic composition will be nullified,
because seasonal variation is impacted it in opposing directions, cancelling out the signal at the yearly
time step. Consequently, if no clear annual trend is observed, it does not mean that recharge is not
changing. Nevertheless, mean annual lake isotopic compositions will be observed to be impacted by
recharge evolution only if it evolves in the same way throughout the year for the most part (i.e.,
consistently decreasing or increasing). In light of these results, it is a monthly time step is strongly
suggested in such investigations, since seasonal recharge fluctuations can be cancelled out in the
yearly signal.

It is also shown that changes in water quality in groundwater-connected lakes depend
substantially on lake location and on the intensity of recharge change. For the studied lakes, in the case
of a +14% recharge increase by 2036, lakes in LAU and OUT regions may experience altered water
quality (driven by phosphorous loading), but no change is expected in the case of a +45% recharge
intensification. If the percentage of recharge increase is at least equal to the percentage of population
growth around the lake, lake quality should not become degraded, but if not, recharge evolution should
be considered in lake management. Lakes water quality in the SAG and AT areas may not decrease
when considering population growth predictions. However, this study does not take into account several
parameters that can greatly impact blue-green algae blooms in lakes, such as the lake water residence
time, chemical threshold processes, and the warming of the water column (Planas and Paquet, 2016).



Finally, even if small groundwater-fed lakes will be sensitive to climate, and especially to
recharge and anthropogenic changes, it is still difficult to predict how their geochemistry will be
impacted, as it is very reactive to each slight variation in water balance parameters. However, more
indicators are now available to predict lake geochemistry evolution, mainly depending on their location
and their G-index. To go further, a recharge model adapted to lake catchments and coupled with melt
dynamics, closely dependent on climate forecasts, could provide more details on lake geochemical
evolution, for more sustainable lake management.



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





Table 1. Main lake characteristics

| Region | ID | Lake name | Lake surface area | Lake volume | Catchment Area |
|--------|----|-----------|-------------------|-------------|----------------|
| | | | $10^3$ m² | $10^3$ m³ | $10^3$ m² |
| AT | 1 | Clair | 115 | 695 | 2646 |
| AT | 2 | Paix | 41 | 97 | 796 |
| AT | 3 | Sauvage | 44 | 142 | 89 |
| OUT | 4 | Lachigan | 33 | 142 | 336 |
| OUT | 5 | Lacroix | 236 | 1080 | 772 |
| OUT | 6 | Lanthier | 25 | 125 | 1134 |
| LAU | 7 | Lacasse | 27 | 67 | 148 |
| SAG | 8 | Beau Portage | 42 | 271 | 364 |
| SAG | 9 | Girard | 67 | 679 | 211 |
| SAG | 10 | Ludovic | 94 | 400 | 1829 |







742 Table 2. Mean isotopic composition of groundwater obtained for the four regions.

| Region | $\delta^{18}O$ | $\delta^2H$ |
|--------|------|------|
| AT | -14.00 | -101.3 |
| OUT | -11.56 | -81.6 |
| LAU | -11.71 | -80.9 |
| SAG | -14.06 | -103.1 |





Table 3. Predicted population growth in the different study regions in 2036 relative to 2011 numbers, according to three different scenarios
(ISQ, 2014)

| Region | Scenarios | | |
|--------|-----------|---|---|
|        | Reference (%) | Low (%) | High (%) |
| OUT | 24 | 13 | 36 |
| AT | 5 | 0 | 10 |
| LAU | 28 | 21 | 34 |
| SAG | 0 | -4 | 4 |

