# Peer review of "Impacts of changes in groundwater recharge on the isotopic composition and # geochemistry of seasonally ice-covered lakes: insights for sustainable"

_Hydrology and Earth System Sciences, 2017_

## Referee Comment (RC1) · Anonymous Referee #1 · 8 Jul 2017

The manuscript presents the results of a model to simulate the isotopic composition of small groundwater-connected lakes in different locations under different future climate scenarios. The approach is not new, however is interesting for the projections and the considerations discussed for future climate scenarios and recharge conditions. In particular, it provides a useful tool for improving our understanding of catchment hydrological processes. Hence, this is a nice work and warrants publication in this journal. However, I have noted a few issues that need to be addressed before the manuscript is considered for publication. Please see my specific comments below

[Figure]

The authors used a lake water budget where the inflow by runoff is considered negligible. This consideration is not explained and there are not geological and hydrogeological description in the paper that could justify this. So, I suggest the author to briefly justify this sentence.

The author assumed a steady state condition for the lakes, it could be under present condition but how it's not clear how this assumption could be true when the authors run future scenario. Under climatic changes and different recharge conditions, are these assumptions satisfied or there is a range in which they could be considered valid? Please, may the authors argument better this part.

In the eq.9 (L231P9) there is the term (–B/V dt), I think that it is not correct because if eq.9 is the solution of eq.8 it means that the eq. 9 is the solution (hence without dt). Please revise or better justify this passage.

The authors do not report the isotopic data, but they say that samples were collected from the top of the epilimnion and from the base of ipolimnion, in case of lake water stratification. But it's not clear what values they use in the model? Average? But in this case for evaporation what values do they use? Please detail this. I suggest also to add a table with isotopic data of lake groundwater and rain water and for Lake Lakasse a figure illustrating the variation of isotopic composition monthly. This could better show the influence of melting periods; hence the authors say that in the 8.5 scenario the isotopic composition would decrease because of melting effect, but in the text, there are not data that support these (or references). Please add data or references.

Do the authors test the sensitivity of the model to investigate the dominant controls on the lake isotope system (a good reference is: Jones et al., 2016. Quaternary Science Reviews, 131:329-340)?

May the authors describe better how they calculate or estimate evaporation (E)?

What values of humidity do the authors use? (ie. from meteorological station?)

[Figure]

The authors repeat in the abstract, in the introduction and in the conclusion that the paper illustrated the effect of future trend on lake geochemistry, but in the paper they discuss only the isotopic composition of water and some consideration about phosphorus load. There are not discussion or results about geochemical data (ie. pH, anions, cations, alkalinity, oxygen dissolved in water…), so I advise the authors to add these data or discussion or to delete the sentence.

In my opinion, the last paragraph about phosphorous is not well connected with the previous part dealing with isotopic model and future scenario. I suggest to link these two parts. Moreover, the phosphorous geochemical behaviour should be different in stratified lake with anoxic water at the bottom. It's not so easy to estimate the quality evolution along different lakes. Do the authors consider the lake geochemistry and thermal/oxygen stratification when they discuss about P load on different lakes?

L183P7: Is the accuracy calculated in relation to deviation of international standard? And wahat are the international standards used? What is the reproducibility?

Is the parameter B (L230P9) m3? I think that is should be a Volume/time.

L271P10: Flake? Is it a typo?

Fig.4: what does the box-whisker describe? (average/median and standard deviation/confidence range/non-outlier min and max?)

L474P20: "…significant relationship…" what does it mean statistically? Do authors perform statistical test? And what?

Fig.9: It's not clear what this figure illustrates. Do they points represent P loads? Is it the results of the model? Please, explain better what the figure wants to describe.

[Figure]

---

## Referee Comment (RC2) · Anonymous Referee #2 · 16 Jul 2017

General comments:

The authors present an interesting study of the variability of the isotopic composition and geochemistry in kettles lakes due to the future variability of recharge and climate. In this aim, the authors compare the measured $\delta 18O$ and $\delta 2H$ in several kettles lakes at annual and monthly intervals and the modeled $\delta 18O$ and $\delta 2H$. The modeled isotopic composition of lake is estimated from climate and estimation recharge models. The modeling results are used to determine if the future evolution of the climate and the recharge could modify the isotopic signature of lake and if the isotopic monitoring in

lakes could be an efficient tool to highlights the variability of water budget and quality. The modeling results have be well analyzed and interpreted, and the authors explain well the assumptions and the limits of their results. The authors study also the water quality but only by the phosphorous. This part, for me, is not really on the topic of this article, less argue than the part about isotopic signature, and maybe not necessary.

Specific comments:

The paper is relatively clear, well written, well structured. Nevertheless, some parts are too long and descriptive and has to modify for a better understanding, notably in the part of results and discussion.

Abstract: The abstract is completed and structured, nevertheless the scientific problematic is not really highlighted, could you add a sentence explaining more clearly the problematic of the paper.

Introduction: Line: 86-88: the interest of this sentence and the link with the end of this paragraph is not clear. Please modify this sentence. The study is based on kettle lakes, this methodological choice should be exposed in the introduction.

Methods: Line 187-190 : the sentence is not clear; please modify it. Water mass balance: several assumptions (Is=0, Ir=0) has not justified, could you please add a sentence to justify this hypothesis. Line 251-254: this sentence is not clear; please modify it. Paragraph evolution scenarios: an introductive sentence could allow a better understanding of this paragraph reminding the interest and using of these models in the study. Line 296-297: Please explain the interest to work with two period, a reference period and future period. Indeed, the reference period is largely in the future. Please explain moreover the choice of 2040 for the transition between these two periods. Figure 2: what represent the dotted line? Fig. 3: It's difficult to understand which model is used, could you clarified this in the caption. In the text, we can suppose that the fig.3a is a result of the publication Arnoux et al., 2017b, if it is the case, could you add the citation in the caption?

Results and discussion:

Monthly evolution of lake isotopic composition Line 373: please, remind quickly how the G-index is measured. Fig. 4: the interest of the close-up is relatively low, without its, the figure will be clearer. Fig. 5, line 389-393: the link between the figure and the interpretation is not clear. We talk about on one hand of reference period on the other hand of the future period while in the figure, the difference between reference period and future period is illustrated.

Annual isotopic signature evolution, isotopic signature evolution. This paragraph is not clear. Indeed, first, line 456-458 the authors explains that lakes with a low G-index and a small volume have higher potential variability in isotopic composition than those with a high G-index and high volume but to illustrate the remark, they used two lakes with a similar mean G-index. Secondly, line 463 to 464, the authors write that "when lakes have a high G-index, the groundwater flux tends to buffer lake isotopic variations, and so they tend to be less sensitive to changes in climate data", but the authors don't give some arguments (results or figure). Please, be clearer. Furthermore, this sentence is not consistent with the figure 8, and the explanation line 476 to 477 " lake isotopic composition is more sensitive to changes in recharge for G-indices ranging from 50 to 80%, with a maximum of sensitivity observed for a G-index of around 65 %. Please clarified this paragraph.

Lake quality evolution This part of the article is disconnected of the other results, where the isotopic variability is analyzed. The scientific interest of the part about the P is really lesser than the rest of the article and not necessary.

Conclusion: This part is clear and well structured. Just, please highltied that when you talk about water,quality you study only the evolution of P. Moreover, the sentence, line 573-575, underlines that the part about P is based on several assumptions (not exposed in the article) and that this part is maybe not necessary on this article.

Technical corrections:

[Figure]

Line 188 : two weeks Line 205: avoid that the ($\delta$p) is not at the same line that precipitation. Line 211: the equation is in subscript. Line 263: two time-levels Line 333: add parenthesis for Rivard et al., 2014, same line 343. Line 364: check the English Figure 6: be careful the indicated period is different between the text and the caption. Line 462: be careful for the reading of the lake volume. Same line 466

---

## Editor Comment (EC1) · B. Schaefli (Editor) · 7 Aug 2017

Both reviewers conclude that this is an interesting manuscript for the readership of HESS and give some detailed comments on how to further improve its readability / methodological details. I would like to invite the authors to respond to these reviews as soon as possible and before preparing the revised version.

---

## Author Comment (AC1) · 15 Aug 2017

Dear Reviewer,

On behalf of all co-authors, I would like to thank you very much for the review of our Manuscript (hess-2017-184), entitled "Impacts of changes in groundwater recharge on the isotopic composition and geochemistry of seasonally ice-covered lakes: insights for sustainable management". Thank you for considering this revision and hoping you will find it suitable for publication in Hydrology and Earth System Sciences. The

[Figure]

Manuscript, revised according to the comments, is attached, as are the responses to the reviewer's comments, in blue in the following.

Thank you,

Sincerely,

Marie Arnoux

Please also note the supplement to this comment:
https://www.hydrol-earth-syst-sci-discuss.net/hess-2017-184/hess-2017-184-AC1-supplement.zip

---

## Author Comment (AC3) · 15 Aug 2017

Dear Editor of Hydrology and Earth System Sciences,

On behalf of all co-authors, I would like to thank you very much for the review of our Manuscript (hess-2017-184-RC1), entitled "Impacts of changes in groundwater recharge on the isotopic composition and geochemistry of seasonally ice-covered lakes: insights for sustainable management". Thank you for considering this revision and hoping you will find it suitable for publication in Hydrology and Earth System Sci-

ences. The responses to the comments have been sent to both reviewers and the Manuscript, revised according to their comments, is attached.

Thank you,

Sincerely,

Marie Arnoux

Please also note the supplement to this comment:
https://www.hydrol-earth-syst-sci-discuss.net/hess-2017-184/hess-2017-184-AC3-supplement.pdf

———————————————————————

[Figure]

**Supplement:**

[revised manuscript text omitted]

---

## Editor Comment (EC2) · B. Schaefli (Editor) · 16 Aug 2017

Reviewer 1 suggested to add some data on isotopes to this paper or a figure illustrating the variation of Lake Lakasse isotopic composition. The authors respond to this saying that all the data is available in other papers of the same author group. I would like to add a comment to this: a scientific paper should be nice to read independant of other papers (as far as possibly); adding some additional information already published elsewhere can be a good choice. This in particular holds if the data has been previously published in journals that are not open access.

---

## Author Response (AR1)

Subject: Review of Manuscript hess-2017-184

Dear Editor of Hydrology and Earth System Sciences,

On behalf of all co-authors, I would like to thank you very much for the review of our Manuscript (hess-2017-184), entitled "Impacts of changes in groundwater recharge on the isotopic composition and geochemistry of seasonally ice-covered lakes: insights for sustainable management". Thank you for considering this revision and hoping you will find it suitable for publication in *Hydrology and Earth System Sciences*.

As suggested, a figure illustrating variations of Lake Lacasse isotopic composition and a table of mean lakes isotopic compositions have been added as supplementary material. The Manuscript, revised according to the comments, is attached, as are the responses to the reviewer's comments, in blue in the following.

Thank you,

Sincerely,

Marie Arnoux

**Reviewer 1:**

The manuscript presents the results of a model to simulate the isotopic composition of small groundwater-connected lakes in different locations under different future climate scenarios. The approach is not new, however is interesting for the projections and the considerations discussed for future climate scenarios and recharge conditions. In particular, it provides a useful tool for improving our understanding of catchment hydrological processes. Hence, this is a nice work and warrants publication in this journal. However, I have noted a few issues that need to be addressed before the manuscript is considered for publication. Please see my specific comments below

We thank Reviewer 1 very much for considering this manuscript, and for all of the helpful comments. Please find our responses to Reviewer 1's comments below.

The authors used a lake water budget where the inflow by runoff is considered negligible. This consideration is not explained and there are not geological and hydrogeological description in the paper that could justify this. So, I suggest the author to briefly justify this sentence.

The chosen kettle lakes do not have any surface stream inflow and are set in fluvioglacial deposits. Overland flow to lakes in the study areas is considered to be low because of the permeable nature of the sandy soils. Moreover, in such a particularly cold continental climate, runoff occurs mainly during the snow melt period as well as groundwater recharge. In previous study on Lacasse lake, we have seen that runoff is negligible face to precipitation and groundwater inflows. Moreover, considering a runoff to kettle lakes negligible has been used by other authors in similar climatic contexts (Isokangas et al., 2015; Krabbenhoft et al., 1990). However we agree with the reviewer that it has to be notified in the text and keep in mind in the uncertainties of the model. Sentences have been added in the method part and in the conclusion for this assumption.

The author assumed a steady state condition for the lakes, it could be under present condition but how it's not clear how this assumption could be true when the authors run future scenario. Under climatic changes and different recharge conditions, are these assumptions satisfied or there is a range in which they could be considered valid? Please, may the authors argument better this part.

A steady state can be considered because, in cold continental climate, lake water level does not vary significantly throughout a year, and water level variation is negligible on the considered yearly time steps. Moreover, considering a steady state lake has been widely used by other authors in such cold continental climates (see, among others, Gibson et al., 2015;Yi et al., 2008;Turner et al., 2010;Kluge et al., 2012;Kluge et al., 2007;Malgrange and Gleeson, 2014).

If we consider a transient state, the balance equations become:

 $\frac{dV}{dt} = I - E - Q = cste \text{ at the considered time step}$ and  $V \frac{d\delta_L}{dt} + \delta_L \frac{dV}{dt} = I\delta_I - E\delta_E - Q\delta_Q$  which gives  $\frac{d\delta_L}{dt} = (I_G\delta_G + P\delta_P - E\delta_E + \delta_L(P + I_G - E - 2cste)) / V$  at the considered time

step.

The use of this equation does not significantly change the results because  $P+I_G-E>>dV/dt$ . In the future, changes in fluxes of the yearly water balance will not be significant enough to modify this because all parameters (P,  $I_G$  and E) should increase between 0 to 50% on a year in considered future conditions (see Figure 2 and Rivard et al. 2014). Moreover, steady state is considered on a monthly time scale for lake Lacasse because it has been already shown that, for this lake  $I_G>>dV/dt$  and this will not change in the future considering water balance parameters predictions (see Arnoux et al., 2017b for more details about lake Lacasse isotopic water balance). Considering these dynamics, we agree with Reviewer 1 that considering steady state impacts the results, but we consider this to be negligible on the considered time step and assume a steady state in the calculations.

In the eq.9 (L231P9) there is the term (-B/V dt), I think that it is not correct because if eq.9 is the solution of eq.8 it means that the eq. 9 is the solution (hence without dt). Please revise or better justify this passage.

Eq 9 is the expression of  $\delta$  evolution in time, depending on  $\delta$  at the time step before and the considered time step. We forgot the time in the A and B terms units, that is why it was probably confusing, units have been modified in the paper (L230 P9).

The authors do not report the isotopic data, but they say that samples were collected from the top of the epilimnion and from the base of ipolimnion, in case of lake water stratification. But it's not clear what values they use in the model? Average? But in this case for evaporation what values do they use? Please detail this. I suggest also to add a table with isotopic data of lake groundwater and rain water and for Lake Lakasse a figure illustrating the variation of isotopic composition monthly. This could better show the influence of melting periods; hence the authors say that in the 8.5 scenario the isotopic composition would decrease because of melting effect, but in the text, there are not data that support these (or references). Please add data or references.

All isotopic data (precipitation, groundwater, lakes) are available in Arnoux et al, 2017a for lakes average values used in the model and in Arnoux et al., 2017b for lake Lacasse monthly values. As suggested, a figure illustrating variations of Lake Lacasse isotopic composition and a table of mean lakes isotopic compositions have been added as supplementary material.

Evaporation used comes from climate models, and  $\delta_E$  is calculated with the isotopic model (cf P8). For RCP 8.5, evolution of temperature, humidity, evaporation and precipitations are illustrated on Fig 2 and show increase in precipitation, evaporation and in temperature more pronounced than RCP 4.5. A description of monthly parameter evolution regarding scenarios and melting effect can be found in Rivard et al. 2014. The text has been improved to better explain from where data used come.

Do the authors test the sensitivity of the model to investigate the dominant controls on the lake isotope system (a good reference is: Jones et al., 2016. Quaternary Science Reviews, 131:329-340)?

Thanks to Reviewer 1 for this interesting reference. Sensitivity analyses has been done on the model in the two references related to the data Arnoux et al, 2017a and b and show that the model is more sensitive to E, h and  $\delta_G$ . A sentence has been added in the method part about this purpose.

May the authors describe better how they calculate or estimate evaporation (E)? What values of humidity do the authors use? (ie. from meteorological station?) Evaporation and humidity come from climate model, as described P10 and 11 and illustrated on Fig 2.

The authors repeat in the abstract, in the introduction and in the conclusion that the paper illustrated the effect of future trend on lake geochemistry, but in the paper they discuss only the isotopic composition of water and some consideration about phosphorus load. There are not discussion or results about geochemical data (ie. pH, anions, cations, alkalinity, oxygen dissolved in water. . .), so I advise the authors to add these data or discussion or to delete the sentence.

We agree with the reviewer 1 that this paper focuses only on a part of lake geochemistry evolution, which are isotopic composition and phosphorous load, and does not treat the complete lake water chemistry which was not the paper aim. As suggested by the reviewer, sentences in abstract, introduction and conclusion have been modified. However, as this paper focuses still on lake geochemistry even if it is a part, we decided to keep the title.

In my opinion, the last paragraph about phosphorous is not well connected with the previous part dealing with isotopic model and future scenario. I suggest to link these two parts. Moreover, the phosphorous geochemical behaviour should be different in stratified lake with anoxic water at the bottom. It's not so easy to estimate the quality evolution along different lakes. Do the authors consider the lake geochemistry and thermal/oxygen stratification when they discuss about P load on different lakes?

We agree with Reviewer 1 that this part is more qualitative than the rest of the paper. How recharge changes can influence P load to lakes is not often taking into account in model studies and we thing that it can be an important aspect to consider. That is why, in this paper, where we talk about how lake geochemistry can change in the future regarding recharge changes, we propose a first estimation of how P load to lake could be affected by recharge change. It is a first step for a more complex model, based on P dynamics in lakes, to determine more precisely how lake will be affect by P load changes in future. Some sentences have been added in this part to better make the link with the rest of the paper and better explain the associated assumptions.

L183P7: Is the accuracy calculated in relation to deviation of international standard? And what are the international standards used? What is the reproducibility?

 $\delta$  values are deviations in per mil (‰) from the isotopic composition of the international standard which is Vienna Standard Mean Ocean Water (VSMOW). The measurement accuracy is  $\pm 1$  ‰ vs VSMOW for  $\delta$ 2H and  $\pm 0.2$  ‰ vs VSMOW for  $\delta$ 18O, considering reproducibility (P7).

Is the parameter B (L230P9) m3? I think that is should be a Volume/time. Thank you to Reviewer 1 for this helpful comment, parameters units have been modified.

L271P10: Flake? Is it a typo?

Flake is the name of the lake model used in the climate simulations (see Mironov et al., 2010;Martynov et al., 2012).

Fig.4: what does the box-whisker describe? (average/median and standard deviation/confidence range/non-outlier min and max?)

The bow-whisker describes median, first and third quartiles and maximum and minimum values, this has been added to Figure legend.

L474P20: "...significant relationship..." what does it mean statistically? Do authors perform statistical test? And what?

The relationship is highlighted by the Figure 8, not by statistical tests; the sentence has been modified regarding this comment.

Fig.9: It's not clear what this figure illustrates. Do they points represent P loads? Is it the results of the model? Please, explain better what the figure wants to describe.

The figure 9 is the result of what is explain in the paragraph and is here to illustrate lakes sensitivity regarding percentage of changes in recharge and in population and therefore in P loads to lakes. The description of the Figure has been improved: The shaded area represents the scenarios for which lakes may be under risk of too high P loading, and therefore at risk of a decrease in water quality. Dots represent lakes in the four study areas for three recharge scenarios.

Reviewer 2:
General comments: The authors present an interesting study of the variability of the isotopic composition and geochemistry in kettles lakes due to the future variability of recharge and climate. In this aim, the authors compare the measured  $\delta$ 18O and  $\delta$ 2H in several kettles lakes at annual and monthly intervals and the modeled  $\delta$ 18O and  $\delta$ 2H. The modeled isotopic composition of lake is estimated from climate and estimation recharge models. The modeling results are used to determine if the future evolution of the climate and the recharge could modify the isotopic signature of lake and if the isotopic monitoring in lakes could be an efficient tool to highlights the variability of water budget and quality.

The modeling results have be well analyzed and interpreted, and the authors explain well the assumptions and the limits of their results. The authors study also the water quality but only by the phosphorous. This part, for me, is not really on the topic of this article, less argue than the part about isotopic signature, and maybe not necessary.

Specific comments: The paper is relatively clear, well written, well structured. Nevertheless, some parts are too long and descriptive and has to modify for a better understanding, notably in the part of results and discussion.

We thank Reviewer 2 very much for considering this manuscript, and for all of the helpful comments. Please find our responses to Reviewer 2's comments below.

Abstract: The abstract is completed and structured, nevertheless the scientific problematic is not really highlighted, could you add a sentence explaining more clearly the problematic of the paper.

Thanks to the Reviewer 2 for this comment, a sentence has been added in the abstract.

Introduction: Line: 86-88: the interest of this sentence and the link with the end of this paragraph is not clear. Please modify this sentence. The study is based on kettle lakes, this methodological choice should be exposed in the introduction.

Thanks to the Reviewer 2 for this comment, the sentence has been modified and a sentence has been added about kettle lakes.

Methods: Line 187-190 : the sentence is not clear; please modify it. The sentence has been modified.

Water mass balance: several assumptions (Is=0, Ir=0) has not justified, could you please add a sentence to justify this hypothesis.

The chosen kettle lakes do not have any surface stream inflow that is why Is=0. Moreover they are set in fluvioglacial deposits, therefore overland flow to lakes in the study areas is considered to be low because of the permeable nature of the sandy soils. Moreover, in such a

particularly cold continental climate, runoff occurs mainly during the snow melt period as well as groundwater recharge. In previous study on Lacasse lake, we have seen that runoff is negligible face to precipitation and groundwater inflows (Arnoux et el. 2017b). Moreover, considering that runoff to kettle lakes is negligible has been used by other authors in such similar climatic contexts (see Isokangas et al 2015; Krabbenhoft et al 1990). However we agree with the reviewer that it has to be notified in the text and keep in mind in the assumption of the model. Sentences have been added in the method part and in the conclusion for this assumption.

Line 251-254: this sentence is not clear; please modify it. The sentence has been modified.

Paragraph evolution scenarios: an introductive sentence could allow a better understanding of this paragraph reminding the interest and using of these models in the study. Sentences have been added.

Line 296-297: Please explain the interest to work with two period, a reference period and future period. Indeed, the reference period is largely in the future. Please explain moreover the choice of 2040 for the transition between these two periods.

This choice has been made because of recharge predictions from Rivards et al. 2014 which are on a reference period, based on actual measurement, and a future 2041-2070 period, therefore to use these data it was necessary to work on a reference period close to present and on 2041-2070 for future period. Also, the reference period has been chosen to cross the two years 2015-2016 field campaign in order to calibrate the model. Furthermore, we decided to choice the same time duration for these two compared periods (to have same signification on means) and the same model for climate data (for the consistency of modelling), that is why we use the 2010-2040 period as the reference period.

Figure 2: what represent the dotted line? The dotted line is just a mark to facilitate the reading.

Fig. 3: It's difficult to understand which model is used, could you clarified this in the caption. In the text, we can suppose that the fig.3a is a result of the publication Arnoux et al., 2017b, if it is the case, could you add the citation in the caption? The Figure 3 caption has been clarified.

Results and discussion: Monthly evolution of lake isotopic composition

Line 373: please, remind quickly how the G-index is measured.

It has been added to the text.

Fig. 4: the interest of the close-up is relatively low, without its, the figure will be clearer.

We agree with the reviewer, however, we decided to keep this representation to show to the reader the range of variations of our results and on what are based the means.

Fig. 5, line 389-393: the link between the figure and the interpretation is not clear. We talk about on one hand of reference period on the other hand of the future period while in the figure, the difference between reference period and future period is illustrated.

As the reference period is the same for all future scenarios (S0, S1, S2 and NC), difference between reference and future with changes ( $\Delta \delta^{18}$ O S0, S1 and S2) can be compared to difference between reference and future with no change ( $\Delta \delta^{18}$ O NC) which is equivalent to a comparison scenarios regarding no change in future. But as suggested by the reviewer, the text has been modified to be clearer.

Annual isotopic signature evolution, isotopic signature evolution.

This paragraph is not clear. Indeed, first, line 456-458 the authors explains that lakes with a low G-index and a small volume have higher potential variability in isotopic composition than those with a high G-index and high volume but to illustrate the remark, they used two lakes with a similar mean G-index. Secondly, line 463 to 464, the authors write that "when lakes have a high G-index, the groundwater flux tends to buffer lake isotopic variations, and so they tend to be less sensitive to changes in climate data", but the authors don't give some arguments (results or figure). Please, be clearer. Furthermore, this sentence is not consistent with the figure 8, and the explanation line 476 to 477 " lake isotopic composition is more sensitive to changes in recharge for G-indices ranging from 50 to 80%, with a maximum of sensitivity observed for a G-index of around 65 %. Please clarified this paragraph.

Thanks to the reviewer 2 for this comment, this paragraph has been clarified: lakes with a low G-index and a small volume have higher potential variability in isotopic composition regarding climate variability (evaporation and precipitation) while lake with G-indices ranging from 50 to 80% have an isotopic composition more sensitive to changes in recharge. We explain first the variability regarding climatic parameters and then, regarding changes in recharge.

**Lake quality evolution**

This part of the article is disconnected of the other results, where the isotopic variability is analyzed. The scientific interest of the part about the P is really lesser than the rest of the article and not necessary.

We agree with the Reviewer 2 that this part is more qualitative than the rest of the paper but we decided to keep it in the paper because how recharge changes can influence P load to lakes is not often taking into account in model studies and we thing that it can be an important aspect to consider. That is why in this paper, where we talk about how lake geochemistry can change in the future regarding recharge changes, we propose a first estimation of how P load to lake could be affected by recharge change. It is a first step for a more complex model, based on P dynamics in lakes, to determine more precisely how lake will be affect by P load changes in future. Some sentences have been added in this part to better make the link with the rest of the paper.

Conclusion: This part is clear and well structured. Just, please highlied that when you talk about water quality you study only the evolution of P. Moreover, the sentence, line 573-575, underlines that the part about P is based on several assumptions (not exposed in the article) and that this part is maybe not necessary on this article.

**The assumptions about lake quality evolution have been added in the lake quality evolution part.**

Technical corrections: Line 188 : two weeks Line 205: avoid that the (δp) is not at the same line that precipitation. Line 211: the equation is in subscript. Line 263: two time-levels Line 333: add parenthesis for Rivard et al., 2014, same line 343. Line 364: check the English Figure 6: be careful the indicated period is different between the text and the caption. Line 462: be careful for the reading of the lake volume. Same line 466 Thanks to the reviewer, the technical corrections have been done.

[revised manuscript text omitted]

---

## Author Response (AR2)

8th September 2017

Subject: Review of Manuscript hess-2017-184

Dear Editor of *Hydrology and Earth System Sciences*,

On behalf of all co-authors, I would like to thank you very much for the review of our Manuscript (hess-2017-184), entitled "Impacts of changes in groundwater recharge on the isotopic composition and geochemistry of seasonally ice-covered lakes: insights for sustainable management". Thank you for considering this revision and hoping you will find it suitable for publication in Hydrology and Earth System Sciences.

We thank you very much for your comments. As suggested, the language has been carefully checked and reread by a native English-speaker. Moreover mathematical notations in the in-line text have been improved. The Manuscript, revised according to the comments, is attached.

Thank you,

Sincerely,

Marie Arnoux